# Δ^9^-Tetrahydrocannabinol (THC): A Critical Overview of Recent Clinical Trials and Suggested Guidelines for Future Research

**DOI:** 10.3390/jcm13061540

**Published:** 2024-03-07

**Authors:** Peter Pressman, A. Wallace Hayes, Julia Hoeng, Diogo A. R. S. Latino, Anatoly Mazurov, Walter K. Schlage, Azhar Rana

**Affiliations:** 1Medicine and Social Sciences, The University of Maine, Orono, ME 04469, USA; 2College of Public Health, University of South Florida, Tampa, FL 33620, USA; awallacehayes@comcast.net; 3Vectura Fertin Pharma, 4058 Basel, Switzerland; julia.hoeng@vecturafertinpharma.com (J.H.); azhar.rana@vectura.com (A.R.); 4Rosa Serra Latino Consulting, 6300 Zug, Switzerland; diogolatino@gmail.com; 5Positum Consulting, Greensboro, NC 27412, USA; 6Independent Researcher, 51429 Bergisch Gladbach, Germany; wk.schlage@t-online.de

**Keywords:** Δ^9^-Tetrahydrocannabinol (THC), clinical trials, clinical research

## Abstract

In this overview, we seek to appraise recent experimental and observational studies investigating THC and its potential role as adjunctive therapy in various medical illnesses. Recent clinical trials are suggestive of the diverse pharmacologic potentials for THC but suffer from small sample sizes, short study duration, failure to address tolerance, little dose variation, ill-defined outcome measures, and failure to identify and/or evaluate confounds, all of which may constitute significant threats to the validity of most trials. However, the existing work underscores the potential therapeutic value of THC and, at the same time, calls attention to the critical need for better-designed protocols to fully explore and demonstrate safety and efficacy. In the most general sense, the present brief review illuminates some intriguing findings about THC, along with the basic threats to the validity of the research that supports those findings. The intent is to highlight existing generic weaknesses in the existing randomized controlled trial literature and, most importantly, provide guidance for improved clinical research.

## 1. Introduction and Background

At least 545 distinct compounds have been isolated from cannabis plants. As with most other complex botanicals, those compounds encompass 20 different classes of chemical compounds, including cannabinoids, terpenes, terpenoids, amino acids, nitrogenous compounds, simple alcohols, aldehydes, ketones, esters, lactones, acids, fatty acids, steroids, non-cannabinoid phenols, pigments, flavonoids, vitamins, proteins, enzymes, glycoproteins, and hydrocarbons. Some of these phytochemicals, including cannabinoids, are concentrated in a resin found in the glandular trichomes of the plant.

At least 125 (Radwan et al., 2021) [1], and perhaps over 140 (Hurgobin et al., 2021) [2], different cannabinoids have been isolated from cannabis. The most studied cannabinoids are THC (Δ^9^-tetrahydrocannabinol), CBD (cannabidiol), and CBN (cannabinol). Others include CBG (cannabigerol), THCA (tetrahydrocannabinolic acid), CBGA (cannabigerolic acid), CBC (cannabichromene), CBDA (cannabidiolic acid), and THCV (tetrahydrocannabivarin). Most cannabinoids exist in two forms: as acids (the most prevalent form in fresh plant tissue) and as neutral (decarboxylated) compounds (typically in processed plant material).

It almost goes without saying that the use of cannabis and cannabinoids recreationally and in the management of medical and psychiatric ailments is widespread globally and historically. We hasten to add a caveat of well-established scientific wisdom about the neurodevelopmental and reproductive toxicity of cannabis exposure in pregnancy, especially in young and reproductive-age females.

## 2. THC

The primary psychoactive and putative medicinal component in cannabis, THC, is formed by the decarboxylation of THCA during the drying step after harvest and mostly after heating (e.g., smoking, vaping or cooking). While THC and cannabidiol (CBD) are the two most studied cannabinoids, CBD, along with combination formulations, appears to be dominating the clinical application landscape. Emerging but incomplete evidence suggests that THC itself may possess some extraordinary potential and may well be under-represented in the research domain.

## 3. Dronabinol

Dronabinol, also known by the trade names Marinol, and Syndros, is a synthetic pharmaceutical form of (−)-trans-Δ^9^-tetrahydrocannabinol, a naturally occurring component of *Cannabis sativa* L. (Marijuana). 

The use of dronabinol in clinical trials is likely appealing to investigators for several reasons: (1) As a synthetic THC, standardization in purity, dose, and formulation is assured. (2) Since dronabinol was approved by the FDA in 1985 (for HIV/AIDS anorexia and chemotherapy-induced nausea and vomiting), it has a long, well-documented history of regulatory compliance, safety, and efficacy. Accepted off-label use includes obstructive sleep apnea.

A synthetic homologue of THC, nabilone, has been recently approved by the FDA but has been available in Canada and Europe for many years. Marketed as Cesamet, nabilone is structurally distinct from THC but mimics its pharmacological activity through weak partial agonist activity at cannabinoid-1 (CB1R) and cannabinoid-2 (CB2R) receptors. Despite its relatively weak affinity for CB receptors, it is considered to be twice as potent as Δ^9^-THC. Though it was approved by the FDA in 1985, the drug only began being marketed in the United States in 2006. It is approved for use in the treatment of anorexia and weight loss in patients with AIDS. Nabilone is a racemate consisting of the (S,S) and the (R,R) isomers (https://pubchem.ncbi.nlm.nih.gov/compound/Nabilone, accessed on 21 December 2023).

## 4. THC Dose–Response Relationship and Variations in Clinical Response

The response to cannabis, in general, and THC, in particular, is often biphasically dose-dependent or hormetic (hormesis; a low-dose stimulation and a high-dose inhibition) with wide individuality in responses. Thus, the same dose and formulation of cannabis may have salutary effects for some but be toxic to others. This observation has led to a call for future clinical protocols to carefully encompass cannabinoid and medical indication-specific dose trials.

Clinical response depends on a multitude of exposure factors (route of administration, duration, and history of use/exposure, frequency of use, and interactions with food and drugs), individual factors (age and gender), and susceptibility factors (genetic polymorphisms of the cannabinoid receptor gene, N-acylethanolamine-hydrolyzing enzymes, THC-metabolizing enzymes, and epigenetic regulations) (Kitdumrongthum et al., 2023) [3].

## 5. THC Classic Toxicology: Absorption, Distribution, Metabolism, and Excretion (ADME) (See Table 1)

The “forensic threshold” of THC that is correlated with decrements in neuropsychological function is known to be between 2 and 5 ng/mL in blood serum for adults. For an appropriately spaced intake of 2 × 2.5 mg THC per day, an adult can be regarded as being at the no observed adverse effect level (NOAEL). Applying a default uncertainty factor of 10 for intraspecies variability to a NOAEL of 2 × 2.5 mg (over ≥6 h) for THC yields a “daily dose of no concern” or a “tolerable upper intake level” of 0.50 mg, corresponding to 7 µg/kg bw. Starting with a NOAEL of 2.5 mg, consumed as a single bolus, the lowest daily acute reference dose (ARfD) of THC would be 0.25 mg, corresponding to 3.5 µg/kg bw for healthy adults, as the most conservative estimate. Other justifiable estimates have ranged up to 14 µg/kg bw per day (Beitzke and Pate, 2021) [4].

**Table 1 jcm-13-01540-t001:** THC absorption, distribution, metabolism, and excretion (ADME) (Huestis et al., 2007) [5].

**Absorption**	
**Bioavailability:**	90–95%, but due to combined effects of first-pass metabolism and high lipid solubility, only 10–20% of the administered dose reaches systemic circulation
**Peak plasma time:**	0.5–4 h (dronabinol and major active metabolite: 11-hydroxy-delta9-THC)
**Peak plasma concentration:**	1.9 ng/mL
**AUC:**	3.8 ng.h/mL
**Distribution**	
**Protein bound**	~97%
**Vd**	10 L/kg
**Metabolism**	
**Metabolites**	Extensive first-pass hepatic metabolism; 11-hydroxy-delta-9-tetrahydrocannabinol (active)
**Elimination**	
**Half-life:**	5.6 h (parent drug); 44–59 h (metabolites)
**Renal clearance:**	18–20 mL/min
**Total body clearance:**	0.2 L/kg/h
**Excretion:**	50% feces; 15% urine
**Pharmacogenomics**	
Systemic clearance of THC may be reduced and concentrations may be increased in the presence of CYP2C9 genetic polymorphism. There is a 2- to 3-fold higher dronabinol exposure in individuals carrying genetic variants associated with diminished CYP2C9 function. Monitoring for increased adverse reactions is recommended in patients known to carry genetic variants associated with diminished CYP2C9 function.	

## 6. Methods: Literature Retrieval

A systematic PubMed search covering the period from April 2013 to June 2023 was conducted by EDANZ (https://www.edanz.com, accessed on 10 November 2023) using the following MESH terms, keywords, and search strings:MESH terms: THC, tetrahydrocannabinol, dronabinol;Keywords: THC, d9-THC, delta-tetrahydrocannabinol, tetrahydrocannabinol, dronabinol;PubMed syntax: (THC, tetrahydrocannabinol, dronabinol [MeSH Terms]) OR (Dronabinol, OR THC OR tetrahydrocannabinol OR *tetrahydrocannabinol) AND (inhale* OR *mucosal* OR oral OR sublingual*).

Relative to administration routes, only oral and inhaled routes of administration were considered. Only clinical trial publications investigating the effect of purified THC were considered. This resulted in a list of articles that were further screened and manually searched for eligible content using the title and the abstract, yielding 24 filtered articles (refer to Table 2). A narrative evaluation of 9 studies that represent the generic limitations and weaknesses of THC trial design is presented and incorporated into the tabulated summary along with the 24 identified trials. The tables organize the trials by clinical indication or diagnosis.

## 7. Results: Narrative Evaluation of Clinical Trials and Methodological Challenges in Study Interpretation [See Table 2a–f for Summaries] 

(**van den Elsen et al., 2017** [14]).

This was a repeated crossover, double-blind, randomized controlled trial (RCT) of 12 weeks duration on the efficacy and safety of twice daily (bid) doses of oral THC in 18 patients with dementia (doses were placebo, 0.75, and 1.5 mg bid). The diagnosis of dementia included “possible” or “probable” Alzheimer’s disease, vascular (multi-infarct) dementia, or mixed dementia. Effects on dementia-associated mobility impairment focused on validated objective and quantitative balance and gait assessments. It was concluded that, in the doses given, THC had a “benign adverse effect profile” and was well tolerated.

Functional impairment is a core symptom of Alzheimer’s disease or any dementing illness and is often measured by a loss of ability to perform activities of daily living. The essential objective of the study was to demonstrate sustained improvement in specific functional capabilities reflected in patients’ mobility status. Diagnostic precision may not be crucial here, but modifiers such as “possible” or “probable” do not convey underlying pathology.

The principal challenge in interpreting trends toward deterioration or improvement during a study of brief duration is teasing intervention effects from the rapid waxing and waning cycle of symptoms. It is noteworthy that Anderson et al. (2017) [28] concluded that for Alzheimer’s patients, the variability in measurements between individuals implies that it will always be difficult to detect even a good treatment effect in a trial of years duration, especially with a small sample size. Additionally, longer durations for trials increase the likelihood of detecting an effect.

Neuropsychological functioning is impacted in a progressive and hierarchical manner associated with cognitive decline, but it has been and remains conventional wisdom among those who study and manage Alzheimer’s disease that substantial variability from one day to the next may remain among the most affected individuals (Arrighi et al., 2013 [29]; Heston, 1997 [30]; Meehan et al., 2002 [31]).

Moreover, individual symptoms evolve differently over the course of dementia. Notwithstanding the general impression that the overall level of neurological and psychopathology increases with dementia severity, the severity level tends to wax and wane, often with severe fluctuation over time. For this reason, longitudinal studies are required to achieve significant insights into the putative post-intervention trajectory of Alzheimer’s disease during the course of the disease (Scassellati et al., 2020 [32]).

Depression and anxiety are integral to the spectrum of dementia, both as a consequence of the disease itself as well as from the patients’ subjective awareness of losses. Following the ideal approach to geriatric research, a rigorous study ought to utilize selected and serially administered depression and mental status measures to attempt to tease out the contributions of tauopathy versus psychosocial variables. The constellation of symptoms often concurs and frequently waxes and wanes in the course of the disease. However, they may serve as prodromal indicators and may influence clinical presentation on any given day. Again, depression and anxiety are risk factors for dementia, but they are not just comorbidities or sequelae. The neurological and psychiatric crosstalk between depression, anxiety, and Alzheimer’s disease generally and ideally requires a long lens or sampling frame to characterize a patient’s baseline and any change in baseline that could be attributed to an intervention (Tanaka et al., 2021 [33]; Tarawneh et al., 2012 [34]).

It is important to note that fluctuating symptoms, even over the course of a day, are not unusual; thus, a dramatic sensorial or motoric improvement may be temporally associated with an intervention but is more likely to be causally linked to the natural history or presentation of Alzheimer’s disease from one day or one week to the next (Donovan et al., 2022 [35]; Mack et al., 2022 [36]; Tarawneh et al., 2012 [34]).

Despite its limitations, Van den Elsen et al.’s study is significant. Abnormal gait is prevalent in established dementia; may discriminate Alzheimer’s from other neurodegenerative disorders; and can also predict progression from normal cognition, cognitive complaints, and mild cognitive impairment (MCI) to dementia syndromes. Quantitative gait parameters, particularly the variability in the stride-to-stride fluctuations, are sensitive markers of neurological dysfunction and are associated with future mobility disability and incident dementia (Downey et al., eds 2017 [37]).

The notion that gait variability may be a putative marker of cognitive–cortical deterioration in neurodegenerative disorders has diagnostic and prognostic implications. In light of data suggesting chronic ultra-low-dose THC is associated with significant augmentation of memory and other aspects of cognition in animal models and in older adults with dementing illness (Bilkei-Gorzo, 2017 [38]; Sarne et al., 2011 [39] and 2018 [40]; Calabrese & Rubio-Casillas, 2018 [41]), the finding that mobility and gait stability are not likely to acutely deteriorate with THC on board represents reassurance about safety. 

(**van Amerongen et al., 2018** [9]).

This “accelerated proof-of-concept study” consisted of 2 phases: a crossover challenge (dose-finding) phase and a 4 week, parallel, randomized placebo-controlled treatment phase. A total of 24 patients (12 in the placebo group and 12 in the treatment group) with progressive multiple sclerosis (MS) and moderate spasticity were enrolled. There were equal numbers of males and females in each group. The aim was to identify any clinical impact and establish an effective dose while minimizing adverse events.

With a combination of objective and subjective measures, i.e., providing convergent validity, it was concluded that the complex interplay of psychoactive effects and analgesia notwithstanding, it appeared that the oral formulation of THC had a stable pharmacokinetic profile, had few adverse events, and appeared to augment the treatment of spasticity and pain associated with MS.

Apart from the very small sample size and brief study duration, an entire constellation of potential confounds was ignored, and this fact is especially salient in the setting of chronic disease, in which personality or coping style is known to have significant impacts on illness behavior and progression; personality traits have long been known to contribute to the phenomenology of disease states and disease progression (Turiano, 2012 [42]; Sutin et al., 2013 [43]).

The way in which patients cope with and express symptoms, especially over the course of chronic disease, is often significantly mediated by a discrete coping style or personality, and this set of variables must be accounted for. Early “Type A” literature linked an aggressive style to cardiovascular outcomes, such as heart attack and stroke (Williams et al., 1980 [44]). More recently, hostility was again suggested as the driving characteristic of the association between “Type A” behavior and cardiovascular disease (MacDougall, Dembroski, Dimsdale, & Hackett, 1985 [45]). Other work demonstrated that specific personality traits are associated with illness severity, duration, and prognosis. For example, higher neuroticism and lower conscientiousness have been found to predict aggregate morbidity (Chapman et al., 2013 [46]) and self-rated health (Turiano et al., 2012 [42]). The same traits are also associated with increased risk for specific diseases (Goodwin & Friedman, 2006 [47]), such as Alzheimer’s disease (Terracino et al., 2014 [48]). Among those living with HIV, disease progression is slower for more open, extraverted, and conscientious individuals (Ironson et al., 2008 [49]). 

(**Carley et al., 2018** [26]).

This was a small, double-blind, randomized placebo-controlled dose escalation pilot trial looking at the effect of two doses of oral dronabinol vs. placebo in a total of 73 patients with moderate to severe obstructive sleep apnea (OSA) over a 6-week study period. A total of 17 patients received a placebo; 19 received 2.5 mg of dronabinol, and 20 patients received 10 mg 1 h before bedtime. This trial is the largest and longest randomized controlled trial of any putative primary drug treatment for OSA. Despite this fact, the authors readily acknowledge that the study likely remains underpowered to detect simultaneous clinically meaningful effects at multiple endpoints. 

(**Bisaga et al., 2015** [17]).

This work represented an 8-week double-blind, randomized, placebo-controlled trial of opiate-dependent subjects in the detoxification and naltrexone induction phases of management. Enrolled subjects had previous cannabis smoking experience. A total of 20 subjects were randomized to placebo, and 40 received 30 mg of dronabinol orally and daily. Outcomes included a subjective assessment of opioid withdrawal with a focus on insomnia and anxiety.

The development of tolerance and the impact on long-term efficacy were not addressed here, and this is a potential threat to the validity of conclusions about clinical utility. Beyond the small sample size and the fact of inter-subject variability, the principal hurdle is the relatively brief duration of the study in the face of a clinical process that typically requires months, if not years, of multi-modal therapies. 

(**Almog et al., 2020** [50]).

This was a randomized, three-arm double-blind, placebo-controlled, crossover trial of inhaled THC via a novel metered-dose device in 24 patients with significant chronic neuropathic or radicular pain or pain associated with chronic regional pain syndrome (CRPS). Apart from the small sample size, the pitfalls of subjective self-assessment, and the preponderance of male subjects, the challenge here is that of chronic pain. Gender, socioeconomic status, educational level, and, most importantly for the experience of pain, coping style or personality traits are potentially significant confounds that, in future studies, should be assessed on some level. Chronic pain is a complex phenomenon that embraces physical, emotional, and social components that are inextricably tied to the somatosensory cortex as well as cognitive and affective processing (Marchand, 2020) [51]. Other potentially significant intervening variables that should be collected and folded into data interpretation include sleep disturbance, fatigue, appetite, and nutritional status. 

(**Weizman et al., 2018** [6]).

This was a randomized, placebo-controlled trial of sublingual THC oil in 15 patients with chronic radicular pain. In addition to looking at the impact of the intervention on the subjective assessment of pain, the authors utilized functional MRI (fMRI) to explore simultaneous neural correlates of cannabis-induced analgesia associated with a single oral dose of THC vs. placebo. What may be most interesting about this work is that putative THC analgesia in chronic pain appears to be mediated through brain areas that underly effective processing of pain as well as supraspinal pain modulation, potentially addressing an imbalance in pain processing dynamics that may occur in chronic pain states.

Women were excluded from this study due to concern regarding menstruation-induced fluctuations in pain sensitivity. In addition, larger-scale studies are mandated not only to assure statistical power but also to look at any generalizable effects on diverse populations. The authors themselves suggested that future investigations should include other related chronic conditions to better understand whether the results represent a pervasive neuronal mechanism of cannabinoid effects on chronic pain or are unique to neuropathic pain states. 

(**Toth et al., 2012** [52]).

This was a 5-week double-blinded trial of 26 subjects with diabetic peripheral neuropathy (DPN) randomized to nabilone or placebo groups (13 placebo and 13 nabilone) in a flexible-dose study. The authors reported that a flexible dose of nabilone 1–4 mg/day orally was effective in relieving DPN symptoms, improving disturbed sleep, and enhancing quality of life. They also subjectively assessed the overall patient status. Nabilone was well tolerated and successful as an adjuvant in patients with DPN, benefiting secondary outcome measures of associated features of the syndrome, including anxiety, depression, sleep efficacy, and subjects’ evaluation of quality of life.

Again, conclusions are weak because the study is underpowered, i.e., it suffers primarily from a small sample size, by the authors’ own admission. Moreover, DPN is a chronic waxing and waning condition, and prolonged treatment with serial assessments over months to years is typically required for definitive results. Dosing of nabilone was performed based upon individual subject tolerability and efficacy, but further dose-finding studies may be more appropriate for determining safety and efficacy if doses higher than 4 mg per day are to be considered. In addition, the authors state that this study was performed to study the adjuvant potential of nabilone in subjects already receiving pharmacotherapeutic management of neuropathic pain from established diabetes; the use of concomitant medications leads to difficulty in teasing out the individual effect of nabilone. Future studies to assess the efficacy of nabilone monotherapy would require a washout of medications, with a focus on the primary efficacy of nabilone for chronic diabetic neuropathic pain. 

(**Grant et al., 2011** [24]).

This was a 12-week uncontrolled pilot study of dronabinol (dose ranging from 2.5 to 15 mg/day) in 12 female patients with a diagnosis of the compulsive hair-pulling disorder, trichotillomania. The authors reported that dronabinol appeared to promote statistically significant reductions in trichotillomania symptoms in the absence of negative constitutional or subjectively assessed cognitive effects. However, larger placebo-controlled studies incorporating validated and widely used cognitive measures are warranted given the small sample and open-label design.

The exclusively female sample may also be a significant confound; the interaction between the gonadal hormones and pain perception is intricate and not well understood. Several clinical pain conditions show significant variation in symptom severity across the menstrual cycle, though there does not appear to be a consensus on whether the menstrual cycle influences experimental pain sensitivity in healthy individuals (Iacovides et al., 2015 [53]). 

(**Malik et al., 2017** [54]).

This was a 28-day pilot study of oral dronabinol (5 mg twice daily) in 13 patients (11 female and 2 male) with non-gastroesophageal reflux and non-cardiac chest pain. These were patients who were diagnosed with odynophagia, hypersensitivity, and pain associated with esophageal dysmotility and who were essentially refractory to mainstream management. The etiology of the challenging symptom picture was attributed to several hypersensitivity mechanisms that include altered autonomic activity, amplified cerebral processing of visceral sensory input, abnormal mechanophysical properties, sustained longitudinal muscle contractions, psychological abnormalities, and increased mucosal mast cells. Improvement was reported in both objective and subjective indices of swallow, esophageal function, and pain. A possible contribution to study outcomes from psychiatric variables was seen as unlikely in light of the results on anxiety and depression inventories.

Threats to validity lie primarily with the small sample size and the brief duration of the study. Assessment of any intervention for broadly defined or vaguely described syndromes that are often diagnoses of exclusion is seen as intrinsically weak. While study subjects appeared to improve with dronabinol, sustained and significant changes should be demonstrated in a larger, appropriately powered sample over a more meaningful study duration and, ideally, in subjects with documented and specific pathologic diagnoses.

## 8. Discussion and Conclusions

In a recent review, Stella (2022 [55]) declared the following: “Considering the wide differences and diversity in the molecular targets engaged by THC [and CBD], phytocannabinoid-based therapeutics will require optimization for each medical indication…” from sleep disorders, anxiety, intractable depression, post-traumatic stress disorder, existential suffering, chronic pain, various malignancies, and dementing illnesses to refractory childhood epilepsy and autism. Indeed, Stella’s suggestion is salient. The large and varied discussion of promising clinical indications for cannabis and cannabinoids, in general, has been reviewed frequently but is far from rigorous, comprehensive, or definitive. Detailed and wide-ranging discussions are intriguing; see Pagano et al. (2022 [56]) and Solmi et al. (2023 [57]). The few randomized clinical trials using THC suggest categories of potential therapeutic opportunities, but these rest on poorly designed clinical trials. To fully explore and operationalize THC as a safe and effective pharmacologic agent, future clinical investigations should encompass the following guidelines:

## 9. Recommendations for Future Studies

Small sample size is a profound problem that plagues many clinical trials. The COVID-19 pandemic certainly played a significant role in making subject recruitment next to impossible for much of the last 2 years. The recent viral pandemic appears to have had much to do with the problem of a small sample size. The sampling and statistical deficiencies in these studies and the need to design and conduct more appropriately powered trials are underpinned by COVID-19. Conducting research during the pandemic required shifting from in-person to virtual recruitment strategies to contact and engage potential study participants, many of whom were very reluctant to participate in studies that they perceived might put them at risk for exposure to the virus. Virtual recruitment for this population during a pandemic was also not at all efficient and hindered efforts to meet recruitment goals (Pertl et al., 2023 [58]).

Moreover, anxieties relating to the pandemic have been elevated, particularly in relation to infection risk due to immunosuppressive treatment, self-isolation, shielding, and difficulty accessing usual care. These concerns exacerbated the baseline reluctance to participate in clinical research (Glintborg et al., 2021 [59]). Future trials will invariably have better success in recruiting and enrolling more meaningful sample sizes.

Study duration is another major factor in definitively drawing conclusions about the impact of any intervention. Even in laboratory animals with very high rates of metabolism, studies looking at interventions for degenerative illnesses have been conducted over 7–12 month periods to evaluate real and enduring effects on neuropsychological and immunohistochemical findings (Kashiwaya et al., 2013 [60]). Meaningful study duration may be up to 2 years to fully and accurately evaluate the impact of a drug or medical food intervention, apart from the normal waxing and waning variations of chronic disease progression.

The study of genetic architectures must be accomplished. Any future study should strive to encompass emerging aspects of the genetic architecture underlying the pathology under investigation. In Alzheimer’s disease, for example, APOE4 is the only common high-risk genetic variant. Genome-wide association studies have further defined common genetic variants, of which about 40 have genome-wide significance. Exome chip analyses have additionally yielded rare variants (SORL1, TREM2, and ABCA7) that strongly increase the risk of early-onset disease (Dai et al., 2017 [61]). Study subjects should ideally be sampled to translate the genetic risk of Alzheimer’s disease into mechanistic insight and nutritional and drug targets for improved clinical management. This is a clear example of the importance of including at least some genotyping in recruiting subjects and then interpreting the results of an intervention with THC; the genetic basis of cannabis use and the variable reactions to cannabinoid administration may be driven and “flavored” by polymorphisms of genes such as *CNR1*, *CADM2*, *FOXP2*, *CHRNA2*, *ANKFN1*, *INTS7*, *PI4K2B*, *CSMD1*, *CST7*, *ACSS1*, *OPRM1*, and *SCN9A*. In exploring the extent to which an individual may respond to a cannabinoid(s), a genuinely comprehensive trial would ideally include a genomic profile of the subjects; with today’s technology, this prospect appears to be realistic and cost-effective (Hillmer et al., 2021 [62]; Verweij et al., 2022 [63]).

The most consistent genetically mediated effects appear to be illuminated in a small body of literature implicating a functional polymorphism within *FAAH* to cannabis dependence as well as withdrawal symptoms and responses to acute THC administration. This evidence is buttressed by a growing neurogenetic literature linking this polymorphism to psychiatrically relevant neuroimaging outcomes, such as threat-related amygdala function and habituation and ventral striatum response to reward, including marijuana cues (Bogdan, 2016 [64]).

Methods of calculating and verifying individual patient doses and developing “personalized/precision medicine” estimates of bioavailability by exposure biomarker measurements should be explored. It is tempting to speculate that this initiative might help to compensate for the varying arrays of metabolic phenotypes and/or drug/drug interactions. Additionally, co-morbidities and previous or ongoing use of opiate analgesics, psychiatric medications, and indeed cannabinoids constitute potential confounds that must be controlled for in well-designed trials.

Objective, validated, and reliable measurements of disease endpoints should ideally be developed to confirm questionnaire outcomes. Surveys, questionnaires, other pencil-and-paper instruments, and digital methods must be selected and administered more carefully to ensure robust but sensitive data.

Gender, socioeconomic status, educational level, and, most importantly, pain experience, coping style, or personality traits should be evaluated in calibrated instrument(s) and validated method(s).

Psychological state and trait and personality measures should be administered at each evaluation visit. Personality or coping style is of significant relevance. Both clinical experience and laboratory work suggest that if personality were an integral part of the study protocol, the data developed would be far more trustworthy.

A recent (Beck et al., 2023 [65]) multi-study project provides robust, conceptually replicated, and extended evidence that psychosocial, i.e., personality factors, are strong predictors of dementia diagnosis and performance on many assessment instruments but not consistently associated with neuropathology at autopsy. The research analyzed data on 44,000 people from eight longitudinal studies, of whom 1700 developed dementia, and compared their personality scores with cognitive test scores and pathology data. Various personality traits, along with educational and occupational status, were significantly associated with performance, i.e., validity and reliability on cognitive and neuropsychological tests, and were indicative of subjects’ capacity to cope with the challenges of neurodegenerative illness. In other words, to fully assess a drug effect, the subjects’ sociodemographic and personality fingerprints should be characterized and evaluated.

Nutritional status should be assessed at the beginning and end of the study. Serum pre-albumin and albumin levels may affect drug bioavailability and dose–response patterns.

Family and medical history should be obtained with meticulous attention to inclusion and exclusion criteria for subject enrollment in a trial.

## Figures and Tables

**Table 2 jcm-13-01540-t002:** (**a**). Clinical studies of purified THC in patients and in healthy volunteers for pain management. (**b**) Clinical studies of purified THC in patients and in healthy volunteers for dementia-related symptoms. (**c**) Clinical studies of purified THC in patients and in healthy volunteers for multiple sclerosis and dementia-related symptoms. (**d**) Clinical studies of purified THC in patients and in healthy volunteers for opioid and cannabis withdrawal. (**e**) Clinical studies of purified THC in patients and in healthy volunteers for stress-related disorders. (**f**) Clinical studies of purified THC in patients and in healthy volunteers for other disorders (hair pulling, metabolic disorders, apnea, etc.). (Efficacy color coding: 
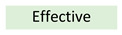


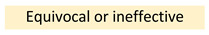


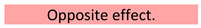
).

(**a**)
**Reference**	**Indication/Condition**	**Delivery/Dose**	**Treatment Duration**	**Number of Patients; Gender Split (M/F)**	**Age (Years)**	**Primary** **Findings**	**Secondary Findings**	**Side Effects**
Weizman et al., 2018 [6]	Chronic lumbar radicular neuropathic pain	Sublingual oils: THC oil or placebo oil (0.2 mg/kg, average THC dose: 15.4 ± 2.2 mg)	2 meetings; in each meeting, patients received THC oil or placebo oil	N = 15 (15:0)	Range: 27–40	Compared with placebo, THC significantly (*p* < 0.05) reduced pain, which was correlated with functional connectivity between the anterior cingulate cortex. Moreover, the degree of reduction was predictive of this response to THC. Graph theory analysis of local measures demonstrated a reduction in network connectivity in areas involved in pain processing and specifically in the dorsolateral prefrontal cortex, which were correlated with individual pain reduction.		
Colwill et al., 2020 [7]	Pain control during medical abortion	Oral: Participants received 800 mg ibuprofen and were randomized to either 5 mg po dronabinol or a placebo 30 min before misoprostol administration	Treatment given once; pain self-report up to 24 h	N = 70 (0:70); dronabinol: 35 (0:35); placebo (0:35)	Dronabinol: 28.1 ± 6.5; placebo: 28.6 ± 5)	No significant difference was found between groups in the median maximum pain score reported at any timepoint (dronabinol 7 [interquartile range 6–8], placebo 7 [interquartile range 5–8]; *p* = 5.85).	Mean maximum anxiety (dronabinol 3.33 ± 3.06), placebo (3.23 ± 2.53), *p* = 5.88; nausea scores (dronabinol 2.21 ± 2.32), placebo 2.72 ± 2.64, *p* = 5.41), side effects (dronabinol 15% (5/33), placebo 6% (2/34); *p* = 5.21) or satisfaction with pain management (76% dronabinol, 82% placebo; *p* = 5.51).	No significant differences between groups. Side effects included vaginal bleeding, muscle pain, headache, nausea, pain in the mouth and throat, paranoia, “giggly”, and increased appetite.
de Vries et al., 2017 [8]	Chronic abdominal pain (i.e., chronic pancreatitis and postsurgical pain)	Oral tablet: tablets with standardized Δ^9^-THC content: Days 1–5, 3 mg tid; Days 6–10, 5 mg tid; Days 11–52, 8 mg tid	Up to 52 treatment days	Chronic = 23 (THC = 8 (7:1); placebo =15 (11:4); postsurgical pain = 27 (THC = 13 (2:11); placebo = 14 (5:9))	Chronic pancreatitis (THC: 53.9 ± 10.3) and postsurgical pain (THC: 52.2 ± 11.3; placebo: 51.9 ± 8.2)	At Days 50–52, VAS mean score did not differ significantly between the THC and placebo groups (*p* = 0.901). Between the start and end of the study, VAS mean scores decreased by 1.6 points (40%) in the THC group compared with 1.9 points (37%) in the placebo group.	No differences were observed in the secondary outcomes.	All (possibly) related adverse events were mild or moderate.
von Amerongen et al., 2018 [9]	Evoked pain	Oromucosal spray: paracetamol (1000 mg), Δ^9^-THC (10 mg), promethazine (50 mg), or matching placebo	Single dose	N = 25 (13:12)	24.0 ± 5.6	Paracetamol was not effective at reducing any of the measured pain modalities. Δ^9^-THC did not show any acute analgesic effect but showed a hyperalgesic effect on two of the five pain tasks, namely, electrical and pressure pain. The negative control, promethazine, showed an increase in pain sensation for cold, pressure, and inflammatory pain.	Subjective alertness, mood, and psychotomimetic symptoms were moderately affected by treatment with Δ^9^-THC (alertness, calmness, and internal and external perception) or promethazine (alertness).	79 TEAEs were registered, of which 54% (*n* = 43) were recorded after treatment with Δ^9^-THC, after which 20 of 25 subjects reported any event.
de Vries et al., 2016 [10]	Chronic abdominal pain in chronic pancreatitis	Oral table: Namisol, 8 mg Δ^9^-THC, or active placebo (5 mg/10 mg diazepam)	Single dose	N = 24 (15:9); (opioid = 12) (8:4); non-opioid = 12 (7:5)	51.8 ± 9.3	No treatment effect was shown for delta VAS pain scores after Δ^9^-THC compared with diazepam.	No significant differences were found between Δ^9^-THC and diazepam for alertness, mood, calmness, or balance. Feeling anxious and heart rate were significantly increased after Δ^9^ THC compared with diazepam.	Δ^9^-THC was generally well tolerated, resulting in only mild to moderate AEs; the most frequently reported AEs after Δ^9^-THC administration were somnolence, dry mouth, dizziness, and euphoric mood.
Schimrigk et al., 2017 [11]	Neuropathic pain	Oral: 7.5–15 mg qd dronabinol; used as adjuvant	48 weeks	N = 240	Range: 21–68	A clinically relevant decrease in mean pain intensities occurred during dronabinol and placebo treatment without reaching statistically significant differences between both groups.		Dizziness, vertigo, fatigue, dry mouth, adverse drug reactions, nausea, headache, diarrhea, and insomnia.
von Amerongen et al., 2017 [11]	Spasticity and neuropathic pain in progressive multiple sclerosis	Oral tablet: placebo and oral formulation of Δ^9^-THC (ECP002A): 3 dose levels: 3, 5, and 8 mg, leading to a total daily dose of 16 mg	4-week treatment phase	N = 24 (8:16); THC = 12 (4:8); placebo = 12 (4:8)	54.3 ±8.9 (THC): 57.3 ± 9.0 placebo: 51.4 ± 8.0)	Pain was significantly reduced when measured directly after THC administration in the clinic but not when measured in a daily dairy. A similar pattern was observed in subjective muscle spasticity.	Other clinical outcomes were not significantly different between active treatment and placebo. Cognitive testing indicated that there was no decline in cognition after 2 or 4 weeks of treatment attributable to THC compared with a placebo.	Nine treatment-emergent adverse events (4.5%) were considered moderate, and one diagnosis (0.5%) of euphoric mood was judged as severe because it led to an inability to work or perform daily activities.
(**b**)
**Reference**	**Indication/Condition**	**Delivery/Dose**	**Treatment Duration**	**Number of Patients; Gender Split (M/F)**	**Age (Years)**	**Primary** **Findings**	**Secondary Findings**	**Side Effects**
van den Elsen et al., 2015a [12]	Dementia-related neuropsychiatric symptoms	Oral tablet: 1.5 mg Namisol or matched placebo	TID for 3 weeks	N = 50 (25:25); placebo = 26 (14:12); THC = 24 (11:13)	78.4 ± 7.4 (THC: 79.0 ± 8.0; placebo: 78.0 ± 7.0	Neuropsychiatric inventory (NPI) total score decreased in both treatment conditions after 14 d (THC, *p* = 0.002; placebo, *p* = 0.002) and 21 d (THC, *p* = 0.003; placebo, *p* = 0.001). There was no statistical difference between THC and placebo over 21 treatment days (change in total NPI: 3.2, 95% CI: 23.6 to 10.0).	No significant differences between the groups in changes to scores for agitation (Cohen–Mansfield Agitation Inventory: 4.6, 95% CI: −3.0 to 12.2), quality of life (Quality of Life Alzheimer’s Disease: −0.5, 95% CI: −2.6 to 1.6), or activities of daily living (Barthel Index: 0.6, 95% CI: −0.8 to 1.9).	No significant differences between groups in the number of patients experiencing mild or moderate adverse events (THC, *n* = 526; placebo, *n* = 514, *p* = 0.36). No effects on vital signs, weight, or episodic memory were observed.
van den Elsen et al., 2015b [13]	Dementia-related neuropsychiatric symptoms	Oral tablets: 0.75 mg Namisol (bid) in blocks 1–3 and 1.5 mg (bid) in blocks 4–6	3 consecutive days	N = 22 (25:7)	76.4 ± 5.3	THC did not reduce NPI compared to placebo (blocks 1–3: 1.8, 97.5% CI: −2.1 to 5.8; blocks 4–6: −2.8, 97.5% CI: −7.4 to 1.8).	No significant differences were found between THC and placebo on agitated behavior and caregiver burden, as measured by PI subscale agitation/aggression, CMA, and ZBI. No differences were found for low-dose THC or high-dose THC vs. placebo on these variables. A substantial increase in CMAJ and ZBI scores was observed over the 12-week study period.	THC was well tolerated, as assessed by adverse event monitoring, vital signs, and mobility. The incidence of adverse events was similar between treatment groups. Four SAEs occurred.
van den Elsen et al., 2017 [14]	Alzheimer’s disease (dementia)	Oral tablets: 3 mg qd (0.05 mg/kg/d) of THC; used as adjuvant	12 weeks	N = 18	Mean = 77	Significantly increased mobility (balance and gait) in patients with dementia.		Similar to placebo.
(**c**)
**Reference**	**Indication/Condition**	**Delivery/Dose**	**Treatment Duration**	**Number of Patients; Gender Split (M/F)**	**Age (Years)**	**Primary** **Findings**	**Secondary Findings**	**Side Effects**
Zajicek et al., 2013 [15]	Primary or secondary progressive multiple sclerosis	Oral capsule: placebo and dronabinol; starting dose: one capsule (3.5 mg Δ^9^-tetrahydrocannabinol equivalent (bid), maximum dose: 28 mg qd	36 months	N = 493 (201:292)Placebo-164 (68:98); dronabinol = 329 (133:196)	52.19 ± 7.8 (dronabinol: 52 ± 7.6; placebo: 51.97 ± 8.2)	145 patients in the dronabinol group had EDSS score progression (0.24 first progression events per patient-year; crude rate) compared with 73 in the placebo group (0.23 first progression events per patient-year; crude rate). HR for prespecified primary analysis was 0.92 (95% CI: 0.68–1.23; *p* = 0.57). The mean yearly change in MSIS-29-PHYS score was 0.62 points (SD 3.29) in the dronabinol group versus 1.03 points (3.74) in the placebo group. Primary analysis with a multilevel model gave an estimated between-group difference (dronabinol–placebo) of −0.9 points (95% CI: −2.0 to 0.2).	Results of multilevel models showed little evidence of an effect of treatment on MSFC, MSWS-12, or RMI.	No serious safety concerns (114 [35%] patients in the dronabinol group had at least one serious adverse event, compared with 46 [28%] in the placebo group).
van Amerongen et al., 2018 [9]	Spasticity and neuropathic pain in progressive multiple sclerosis	Oral tablet: placebo and oral formulation of Δ^9^-THC (ECP002A): 3 dose levels: 3, 5, and 8 mg, leading to a total daily dose of 16 mg	4-week treatment phase	N = 24 (8:16); THC = 12 (4:8); placebo = 12 (4:8)	54.3 ± 8.9 (THC: 57.3 ± 9.0; placebo: 51.4 ± 8.0)	Pain was significantly reduced when measured directly after THC administration in the clinic but not when measured in a daily diary. A similar pattern was observed in subjective muscle spasticity.	Other clinical outcomes were not significantly different between active treatment and placebo. Cognitive testing indicated that there was no decline in cognition after 2 or 4 weeks of treatment attributable to THC compared with placebo.	Nine treatment-emergent adverse events (4.5%) were considered moderate, and 1 diagnosis (0.5%) of euphoric mood was judged as severe because it led to an inability to work or perform daily activities.
Ball et al., 2015 [16]	Progression in multiple sclerosis	Oral capsule: oral Δ^9^-THC (maximum 28 mg/day) or matching placebo	3 years	N = 493 (201:292); THC = 329 (133:196); placebo = 164 (68:96)	52.19 ± 7.8 (THC: 52.29 ± 7.6; placebo: 51.97 ± 8.2)	No significant treatment effect: hazard ratio EDSS score progression (active: placebo) 0.92 [95% confidence interval (CI): 0.68 to 1.23]; estimated between-group difference in MSIS-29phys score (active: placebo) –0.9 points (95% CI: –2.0 to 0.2 points).	No significant treatment effects. There was no clear symptomatic or disease-modifying treatment effect. The estimated mean incremental cost to the NHS over usual care over 3 years was GBP 27,443.20 per patient. There were no between-group differences in QALYs.	At least one SAE: 35% and 28% of active and placebo patients, respectively.
(**d**)
**Reference**	**Indication/Condition**	**Delivery/Dose**	**Treatment Duration**	**Number of Patients; Gender Split (M/F)**	**Age (Years)**	**Primary** **Findings**	**Secondary Findings**	**Side Effects**
Bisaga et al., 2015 [17]	Opioid withdrawal	Oral capsule: 30 mg/d (0.5 mg/kg/d) of dronabinol; used as adjuvant	8 weeks	N = 40	Range: 18–60	Reduced the severity of symptoms during acute impatient detoxification.		Insomnia, mood changes, fatigue, diarrhea, increased/decreased appetite, nausea, gastrointestinal distress, and sweating.
Jicha et al., 2015 [18]	Opioid withdrawal	Oral capsule: 5–40 mg/day (0.08–0.6 mg/kg/d) of dronabinol; used as monotherapy	5 weeks	N = 20	Range: 18–50	Poorly tolerated (40 mg), better tolerated (20–30 mg), placebo-like effects (5–10 mg).		Dose-related; sustained sinus tachycardia and anxiety (*n* = 3).
Lofwall et al., 2016 [19]	Opioid withdrawal	Oral capsule: 5–40 mg/day (0.2–0.6 mg/kg/d) of dronabinol; used as adjuvant.	5 weeks	N = 20	Range: 18–50	Modest evidence of withdrawal suppression effects for a limited duration (3.5–4.5 h) after dosing 20–30 mg; not a likely monotherapy candidate.		High sedation, bad effects, tachycardia, anxiety, and panic.
Levin et al., 2015 [20]	Cannabis withdrawal	Oral: 60 mg/d of dronabinol used as adjuvant	11 weeks	N = 156	Mean: 35 years	No significant effects as a treatment for cannabis use disorder with lafutidine.		Dry mouth, intoxication, anxiety, and hypotension.
(**e**)
**Reference**	**Indication/Condition**	**Delivery/Dose**	**Treatment Duration**	**Number of Patients; Gender Split (M/F)**	**Age (Years)**	**Primary** **Findings**	**Secondary Findings**	**Side Effects**
Childs et al., 2017 [21]	Acute psychosocial stress	Oral capsule: one capsule per session: 0, 7.5, or 12.5 mg of THC	Two 4 h sessions: one with a psychosocial stress task and one with a non-stressful task (control); sessions were 5 days apart; same dose at both sessions	N = 42; 0 mg THC = 13 (9:5); 7.5 mg THC = 14 (9:6); 12.5 mg THC = 15 (11:2))	23.6 ± 0.7	In comparison with placebo, 7.5 mg THC significantly reduced self-reported subjective distress after the TSST and attenuated post-task appraisals of the TSST as threatening and challenging. By contrast, 12.5 mg THC increased negative mood overall, i.e., both before and throughout the tasks, and pre-task ratings of the TSST were threatening and challenging.	12.5 mg THC impaired TSST performance and attenuated blood pressure reactivity to the stressor. In comparison with placebo, THC did not dose-dependently alter MAP or salivary cortisol during the 2 h pre-treatment period. There was a trend toward THC-induced heart rate elevation [Group × Time F(4,78) = 2.3 *p* < 0.07 ηρ^2^ = 0.11] and analysis of change scores at time point 3 showed a significant effect of 7.5 mg THC upon heart rate [Group F(2,41) = 4.2 *p* < 0.05].	Not indicated
Roepke et al., 2023 [22]	Nightmares in post-traumatic stress disorder. (study protocol)	Oral oil: dronabinol (BX-1; 25 mg/mL dronabinol) or placebo: once-daily oral dose before bedtime	10 weeks	N = 176 (targeted); individual dose titration	18–65 years (targeted)	Primary outcome measure: frequency and intensity of nightmares, measured with the Clinician-Administered PTSD Scale-IV (CAPS-IV) B2 score for the last week, range 0–8. A lower score indicates less frequent and/or intense nightmares.	20 additional scores/self-reported outcomes as secondary outcomes).	not indicated
Zabik et al., 2023 [23]	Extinction learning and fear renewal in post-traumatic stress disorder	Oral capsule: acute oral dose of THC: 7.5 mg of dronabinol	Single dose	THC = 34 (17:17); placebo = 37 (19:18)	THC: 26.5 ± 5.6; placebo: 25.8 ± 6.1	During early extinction learning, individuals with PTSD given THC had greater vm PFC activation than their TEC counterparts. During a test of the return of fear (i.e., renewal), HC and individuals with PTSD given THC had greater vm PFC activation compared to TEC. Individuals with PTSD given THC also had greater amygdala activation compared with those given PBO. We found no effects of trauma group or THC on behavioral fear indices during extinction learning, recall, and fear renewal.		
(**f**)
**Reference**	**Indication/Condition**	**Delivery/Dose**	**Treatment Duration**	**Number of Patients; Gender Split (M/F)**	**Age (Years)**	**Primary** **Findings**	**Secondary Findings**	**Side Effects**
Grant et al., 2022 [24]	Body-focused repetitive behaviors (hair pulling and skin picking)	Oral capsule: started at 5 mg/day of dronabinol for 2 weeks, then 5 mg twice a day for 2 weeks, and then 5 mg three times per day for the remaining 6 weeks	10 weeks	N = 50; placebo = 25 (6:19); dronabinol = 25 (3:21)	Placebo: 28.36 ± 7.27 dronabinol: 33.04 ± 12.48	Dronabinol and placebo treatment were associated with significant reductions in BFRB symptoms, but there were no significant differences between the groups.	At week 10, 67% of the treatment group were classified as responders (Clinical Global Impressions Improvement Score of very much or much improved) compared to 50% in the placebo group (*p* value = 0.459).	Dronabinol was associated with more frequent side effects than placebo, but AEs were generally mild to moderate in intensity.
Reichenbach et al., 2015 [25]	Metabolic disorder	Oral capsule: 10 mg/d (0.2 mg/kg/d) of dronabinol; used as monotherapy	4 weeks	N = 19	Range: 18–75	No significant effects on metabolic parameters (BMI, HDL, triglycerides, LDL, insulin, leptin, AST, ALT, LDH, glucose, and high-sensitivity C-reactive protein).	-	None.
Carley et al., 2018 [26]	Obstructive sleep apnea	Oral capsule: 2.5–10 mg/d of dronabinol (0.04–0.2 mg/kg/d)	6 weeks	N = 73	Range: 21–65	Significantly reduced the apnea–hypopnea index, improved self-reported daytime sleepiness, and greater overall treatment satisfaction.	-	Sleepiness, drowsiness, headache, nausea, vomiting, dizziness, and lightheadedness.
Dunn et al., 2021 [27]	Analgesia, abuse liability, and cognitive performance	Oral capsule: combinations of placebo, hydromorphone (4 mg; oral), and dronabinol (2.5, 5.0, and 10 mg; oral)	Study drugs were co-administered at 10:00 a.m. each session; five outpatient laboratory sessions were scheduled a minimum of 7 days apart	N = 29 (14:15)	30.4 ± 9.2	A consistent dose–effect relationship of dronabinol on hydromorphone across all measures was not observed. Analgesia only improved in the hydromorphone + dronabinol 2.5 mg condition. Hydromorphone + dronabinol 2.5 mg showed the lowest risk and hydromorphone + dronabinol 5 mg showed the highest risk for abuse. Hydromorphone + dronabinol 10 mg produced a high rate of dysphoric effects. Overall, only hydromorphone + dronabinol 2.5 mg modestly enhanced hydromorphone-based analgesia, and hydromorphone + dronabinol 5 mg and 10 mg increased the risk for abuse and AEs.	Subgroup analyses showed subjective effects and abuse risk increased among opioid responders and were largely absent among non-responders.	Hydromorphone + dronabinol 5 mg and hydromorphone + dronabinol 10 mg produced AEs.

## Data Availability

No new data were created or analyzed in this study. Data sharing is not applicable to this article.

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
