# Peer review of "Δ9-Tetrahydrocannabinol (THC): A Critical Overview of Recent Clinical Trials and Suggested Guidelines for Future Research"

_jcm, 2024, doi:10.3390/jcm13061540_

Round 1
Reviewer 1 Report
Comments and Suggestions for Authors
It is a well-structured comprehensive review paper focusing on THC and its potential role as adjunctive therapy in various medical illnesses. More specifically, the authors describe
the existing work showing the potential therapeutic value of THC and highlight existing generic weaknesses in the existing randomized controlled trial literature to provide in turn suggestions to improve clinical research. This topic is very important, and in light of its clinical implications this review is timing and helpful. There are few issues that should be addressed by the authors in order to increase the impact of this review, including the addition of missing information.
Minor concerns:
The studies which are evaluated and analysed in this manuscript are focused on the potential therapeutic effects of adult THC treatment in several pathologies. Given thegrowing liberalization of Cannabis use across the globe, I strongly suggest to add in the section “discussion and conclusion” some limits to the THC use during sensitive periods of the life such as the pregnancy and the adolescent age. More specifically, several preclinical and human studies (see: 1] Drazanova et al., 2019 Scientific Reports 9:6062. doi: 10.1038/s41598-019-42532-z. 2] Di Bartolomeo et al., 2021 Pharmacol Res. 164:105357. doi: 10.1016/j.phrs.2020.105357 3] Di Bartolomeo et al., 2023 Int J Mol Sci. 24:3907. doi: 10.3390/ijms24043907; 4] Stark et al., 2021 Pharmacol Res. 174:105938. doi: 10.1016/j.phrs.2021.105938, which should be cited) shown that gestational/adolescent THC exposure may facilitate the development of psychopathologies at adulthood. Thus, it should be also interesting to discuss this aspect.
Furthermore, it is well known that the cannabinoid CB1 receptors, the THC main target, are present at very high levels on inhibitory (GABAergic interneurons) and at a lesser extent on excitatory (glutamatergic) terminals (Marsicano and Lutz, 1999), as well as on neurons expressing dopamine D1 receptors, playing a specific role in the repertoire of different emotional behaviours. Thus, the authors cannot exclude that the THC effects could be due to specific involvement of CB1 receptors expressed on different neuronal subpopulations as suggested by the following papers, which should be cited.
a) Ruat et al., 2021. Genes Brain Behav. 2021 Nov;20(8):e12775. doi: 10.1111/gbb.12775.
b) Terzian et al., 2014 EJN doi: 10.1111/ejn.12561.
c) Steiner et al., 2008 Psychoneuroendocrinology.33(8):1165-70. doi: 10.1016/j.psyneuen.2008.06.004.
d) Micale et al., 2017, J. Psych. Res. doi: 10.1016/j.jpsychires.2017.02.002.
Conclusions:
I feel that minor revision is required, especially regarding the above citations to add in the text.
Author Response
Responses to Reviewer 1
Regardless of recreational or medical classes of use, it is advised that we add cautions and caveats about the well-demonstrated neurodevelopmental and reproductive toxicity, especially during pregnancy and in otherwise healthy reproductive age females and military age males.
In a general sense, we of course, very much appreciate and concur with this suggestion. We have added the important caution (now, essentially conventional wisdom) as suggested, but respectfully, since the focus of our manuscript is strictly on abuse and dependence along with the strengths and weaknesses of the associated clinical research, we’ll elect to limit the length of the qualifying comment and choose not to expand the references …which we see as beyond the scope of the review and commentary.
It is suggested that we expand the discussion of the complex receptor biology that underlies the pharmacology of THC. While we believe that the suggested discussion would be fascinating, it again seems well beyond the scope of the present exercise. We intend to review the pharmacology in depth in a follow-on manuscript.
Reviewer 2 Report
Comments and Suggestions for Authors
This is an important and timely review on clinical trials using THC for treatment of psychiatric and neurological disorders. The paper is well written and all studies are critically evaluated. I have one issue which is treatment of THC for pain. It is important to assess whether patients were previously treated with opioid based medication. It is also important to compare the effects of THC and opioid analgesic medication in terms of success and side effects. Secondly, from the descriptions, some of the studies don't show any clear results and it is unclear how they were measured- for example Van den Elsen (2017). The general problem of clinical studies using THC is that they are mostly not placebo-controlled clinical trials but preliminary studies with a small sample size and short duration, as this review has pointed out.
Author Response
Responses to Reviewer 2
We very much appreciate the kind and insightful comments of Reviewer 2. It is observed that data on previous use of opioid (as well as cannabinoid) medication is notably absent from most existing clinical trials; this is an important deficiency across most trials at this point, and in discussing the failure to control for confounds such as previous use of potentially addictive or psychiatric meds, we have added a sentence underscoring this weakness.
Additional detail on efficacy and side effects is suggested. With respect, and as noted previously, we feel strongly that additional detail on these areas in an already lengthy otherwise focused review and commentary would only serve to dilute the conclusions and recommendations arising from our analysis of existing trials.
Response to Editor's Note
Based upon a commercial software screen, there was concern about some language declared as redundant with that of existing research and review literature. We are emphatic about meticulous citation of all sources; the software overwhelmingly highlighted phrases that are standardized terms of art and represented precisely stated summaries of cited trials, thus yielding what amounts to a false positive for use of "redundant language." We strongly assert that our thesis in the present ms is a fundamentally provocative and original position that required extensive and accurate quotation and citation of existing evidence in order to present our perspective.